# Injection-Molded Isotactic Polypropylene Colored with Green Transparent and Opaque Pigments

**DOI:** 10.3390/ijms24129924

**Published:** 2023-06-08

**Authors:** Vaclav Janostik, Vojtech Senkerik, Lukas Manas, Michal Stanek, Martin Cvek

**Affiliations:** 1Faculty of Technology, Tomas Bata University in Zlin, Vavreckova 5669, 760 01 Zlin, Czech Republic; vsenkerik@utb.cz (V.S.); lmanas@utb.cz (L.M.); stanek@utb.cz (M.S.); 2Centre of Polymer Systems, University Institute, Tomas Bata University in Zlin, Trida T. Bati 5678, 760 01 Zlin, Czech Republic

**Keywords:** polypropylene, injection molding, mechanical testing, pigments

## Abstract

Polypropylene (PP) belongs among the most important commodity plastics due to its widespread application. The color of the PP products can be achieved by the addition of pigments, which can dramatically affect its material characteristics. To maintain product consistency (dimensional, mechanical, and optical), knowledge of these implications is of great importance. This study investigates the effect of transparent/opaque green masterbatches (MBs) and their concentration on the physico-mechanical and optical properties of PP produced by injection molding. The results showed that selected pigments had different nucleating abilities, affecting the dimensional stability and crystallinity of the product. The rheological properties of pigmented PP melts were affected as well. Mechanical testing showed that the presence of both pigments increased the tensile strength and Young’s modulus, while the elongation at break was significantly increased only for the opaque MB. The impact toughness of colored PP with both MBs remained similar to that of neat PP. The optical properties were well controlled by the dosing of MBs, and were further related to the RAL color standards, as demonstrated by CIE color space analysis. Finally, the selection of appropriate pigments for PP should be considered, especially in areas where dimensional and color stability, as well as product safety, are highly important.

## 1. Introduction

Polypropylene (PP) is a widely used thermoplastic polymer exhibiting excellent properties demanded in a variety of applications. It is a semi-crystalline polymer, and its morphology is affected by the crystallization process. PP crystallizes through numerous processes, which lead to the development of complex crystalline morphologies in the produced elements. The nucleation process can be initiated by spontaneous, heterogeneous, or flow-induced nucleation. Heterogeneous nucleation involves the creation of crystalline cores on the surface of foreign particles or by the addition of a nucleation agent [1,2].

PP elements are often required to have a desired color, which can be obtained by the addition of small amounts of coloring concentrates composed of dyes and pigments, which may have a significant impact on the nucleation processes. There have been several studies focused on the influence of various pigments on the properties of PP. For instance, the extruded PP fibers colored with quinacridone and phthalocyanine pigments showed the formation of *α* and *β* modifications. The extent of these polymorphic modifications depended on the type/concentration of pigment and processing parameters, such as the extrusion rate and temperature [3]. The nucleating abilities of quinacridone and phthalocyanine pigments were also investigated by differential scanning calorimetry (DSC) and polarized microscopy, revealing a slightly higher efficiency of the phthalocyanines [4]. The other study compared the properties of extruded PP colored with liquid color concentrates (LCCs) and pigmented MB [5]. It was found that both types of colorants were similarly effective in terms of their mechanical, rheological, and optical properties. Interestingly, no changes in crystallinity were observed between PP containing LCC or MB. All additives that increase the frequency of polymer nucleation result in the development of internal stresses and changes in the polymer structure [1], which cause undesirable straining and shrinking of the final colored PP products. Since papers on colored PP are primarily focused on the PP in the form of extruded fibers, the shrinkage phenomenon is usually overlooked. However, this adverse phenomenon is particularly important in injection molding technology, where strict dimensional requirements are imposed on the final product.

Shrinkage is defined as a relative difference between the dimensions of the mold cavity and the final part. The primary shrinkage is measured between 16 and 24 h after manufacturing and represents around 90% of the entire shrinkage since the dimensions are not fully stabilized. Secondary shrinkage can occur up to 1000 h after manufacturing and represents an issue, especially with semi-crystalline materials, which undergo a transition from an amorphous to a semi-crystalline phase when stored or used above their glass transition temperature. In the industry, complete shrinkage is most commonly measured, encompassing both the manufacturing (primary) and the subsequent (secondary) shrinkage. Final shrinkage is usually measured at a given time after manufacturing. The interval is given by the customer or an appropriate standard [6,7,8]. In order to describe the effects occurring during the contracting phase, pVT diagrams are used, including pressure (p), volume (V), and temperature (T) [9,10]. In this regard, several studies were conducted to examine the possibility of decreasing shrinkage in the injected samples [11,12,13,14].

Injection molding is a cyclical thermodynamic shaping process that allows the use of a multi-cavity injection mold and results in mass production in one cycle [15,16,17]. The effect of injection molding process parameters on shrinkage values was tested on various polymers, including acrylonitrile butadiene styrene (ABS), polystyrene (PS), and high-density polyethylene (HDPE) [18]. The measurements were evaluated by the Taguchi method, which revealed that the semi-crystalline polymer shrank more in a direction perpendicular to the melt flow than it did in the flow direction. However, the main parameters affecting the value of shrinkage were the mold and melt temperatures.

Additionally, the effect of various process parameters on the volumetric shrinkage of two types of polymers, i.e., semi-crystalline HDPE and amorphous polycarbonate (PC), showed that semi-crystalline materials demonstrate higher values of shrinkage than their amorphous counterparts. The results also showed that the flow path is the dominant factor influencing the final shrinkage [18]. On the other hand, technological parameters analyzed by the ANOVA method revealed that the most important parameter influencing shrinkage is the cooling time [19]. The other study [2] focused on shrinkage evaluation during the duration of holding pressure, the injection itself, and immediately after the mold opening. The beginning of the shrinkage process was observed before the mold opened. This method was carried out with strain gauges placed within the mold. The experimental results implied that shrinkage decreases with increasing pressure and the duration of the holding pressure phase. The length of the holding pressure phase influences the shrinkage up until the gate freezes. The results gained from the strain gauges showed that the start of the shrinking process can be delayed by increased holding pressure.

The aim of this research was to complexly investigate the behavior of the injection-molded PP upon introducing two different green MBs in various concentrations. The first MB contained organic dyes, such as phthalocyanines, resulting in a transparent green PP. The second MB was based on mostly inorganic pigments, such as chromium and copper oxides, resulting in opaque green products. Our study began with the design, optimization, and fabrication of a laboratory injection mold to produce the test specimens. The implications of pigments on the dimensional stability, crystallinity, and morphology were investigated and correlated with the changes in the rheological, mechanical, and optical properties of iPP. The results demonstrate that even small amounts of pigment significantly affect the final properties, which should be taken into consideration for engineering and safety reasons when utilizing such materials in real-world applications.

## 2. Results and Discussion

Since we aimed to complexly investigate the effect of the colorants on the behavior of the isotactic PP, the standardized injection mold was designed (Figure 1a) and fabricated (EN ISO 294-4), and its dimension accuracy was verified using the 3D optical scanner. As can be seen in Figure 1b, the laboratory mold was successfully produced within the appropriate tolerances (±0.1 mm) and was deemed applicable for the fabrication of the representative test bodies. Subsequently, the injection molding process was simulated using Autodesk Moldflow (version 2016) software (Autodesk, San Francisco, CA, USA) to provide better insight into the cavity filling and to facilitate the determination of the processing parameters for the sample fabrication. The simulation showed a uniform melt velocity profile and a balanced filling of the mold cavity (Figure 1c), which further verified the correct mold design. The theoretical filling time of the cavity was estimated to be ~2.5 s, while the theoretical shrinkage of the sample was around 0.98%. It should be noted that the theoretical shrinkage was identical in the flow and transverse directions, even for the neat PP, which was later disproved by the experiments. The simulation was performed for the neat PP and could not account for the influence of the color MBs. Using the optimized processing conditions, the representative samples were fabricated (Figure 1d) and tested for (i) shrinkage, (ii) crystalline content, (iii) microstructural/elemental analysis, and (iv) rheological, (v) mechanical, and (vi) optical properties.

### 2.1. Shrinkage Evaluation

For each determined concentration, 15 specimens were fabricated and tested for shrinkage. Figure 2a,b displays the mean values of shrinkage for both the t-PP and the o-PP, collected in the flow as well as the transverse direction. The results clearly show that the presence of the PP-based coloring substances (PP616297 and PP620577/12) within PP influenced the shrinkage of the material. In each case, the shrinkage remarkably increased, which could be explained by changes at the molecular level that are induced by the presence of pigments [2,13,20]. As is known, pigments can act as nucleation agents in semi-crystalline polymers, influencing their structure [2,21]. The shrinkage of the t-PP was less pronounced compared to its o-PP analog, which can be explained by the different size of the pigments (elaborated further in the text), resulting in a higher ordering of molecular chains that directly increases the shrinkage [22,23]. Upon a major increase (from 0 to 1 wt%), the shrinkage further increased nearly linearly with the MB concentration, within the investigated concentration range (1–5 wt%).

It is obvious that shrinkage slightly varied in both the flow direction and the transverse direction; however, the type of the coloring MB caused a major difference in the evolution of the shrinkage phenomenon. The relative increments of the shrinkage compared to the neat PP are displayed in Figure 2c,d. As can be seen, the relative shrinkage in the main flow direction increased from 13.6 to 22.3%, and from 49.5 to 58.3%, for the t-PP and o-PP, respectively, when increasing the MB content from 1 to 5 wt%. While t-PP showed similar shrinkage values in the main flow as well as the transverse direction, o-PP exhibited slightly lower relative shrinkage values in the transverse direction (from 39.4 to 51.5%), most probably due to the preferential orientation of the opaque pigments during the injection event [1,22]. The shrinkage measurements suggested that the addition of the coloring MB supports the growth of crystals within the PP matrix.

### 2.2. Crystallinity Evaluation

The XRD analysis was performed to analyze the content of the crystalline phase in the injection-molded samples. The neat PP served as a reference to identify characteristic XRD peaks. The diffraction peaks pertaining to the crystallographic planes (110), (040), (130), (111), (131), and (011) matched the standard JCPDS Card No.: 66-1214, corresponding to *α*-isotactic PP. Moreover, the characteristic (300) plane referring to *β*-type crystals was not detected [24]. Figure 3a compares the XRD patterns of the reference with the t-PP, o-PP analogs. The degree of crystallinity, *χ_C_*, was determined using the following Equation (1):(1)$$\chi_{C}=\frac{A_{\mathrm{crystalline}}}{A_{\mathrm{crystalline}}+A_{\mathrm{amorphous}}}\times 100\%$$
where A_crystalline_ and A_amorhous_ represent the corresponding areas of specific regions in the XRD spectrum [25]. The calculated *χ_C_* values and relative increase in *χ_C_* are shown in Figure 3b. The results demonstrate that *χ_C_* gradually increased with the content of each coloring MB; applying transparent pigments to the iPP matrix generally resulted in a less pronounced growth of crystals when compared to opaque pigments, e.g., an increment of up to 7.6% was detected in t-PP-5, while an increment of up to 10.6% was found in o-PP-5.

In order to obtain transparent coloring in a polymer, it is necessary to incorporate pigments of very small dimensions, generally smaller than the differentiating ability of a human eye [14]. Therefore, it was assumed that transparent (nano)particulate pigments would act as nucleation origin points, affecting the growth of the crystal to a greater extent compared to opaque pigments with presumably larger dimensions. Nevertheless, the opposite trend was observed (Figure 3b), which could be explained as follows: (i) the total *χ_C_* value includes the additional crystallinity given by the inorganic pigment, and/or (ii) different surface properties/chemistries of the pigments affected the lamellae formation to a different degree [23]. The mechanical analysis supported the assumption of a different particle/matrix interface, as shown later (Section 3.5). Moreover, the XRD data agree with the shrinkage phenomenon (Figure 2), which supports the validity of the observed trend.

### 2.3. Microstructure Analysis

The microstructure and the elemental composition were investigated using SEM/EDX. For brevity, we present only samples with the highest concentration of each MB, i.e., t-PP-5 and o-PP-5. As seen in Figure 4, the surface of the samples (in contact with mold) was relatively smooth without any imperfections; a subtle protrusion of the inorganic pigments was detected. The presence of oval-shaped/spherical pigments was successfully identified, despite their small size and low concentration. Pigments embedded in transparent PP were smaller when compared to those in the opaque analog, and were sized below the differentiating ability of a human eye (Figure 1). The pigments in the opaque PP also had submicron dimensions, with a reasonable compatibility with the PP matrix. They were evenly distributed without any occurrence of aggregation, which proves the success of the fabrication method.

EDX analysis was used to determine the elemental composition of the samples (Figure 4). As expected, the neat PP contained mostly C and O; the presence of Au/Pd originated from the sputtered layer. In the case of PP supplemented with the green colorants, both EDX spectra showed a decent signal for Cr and Cu, while the spectrum for o-PP-5 also exhibited a minor peak for Ti. These elements proved the presence of coloring agents represented by copper phthalocyanine [26], chromium oxide, and titanium oxide [27,28]. In addition, the EDX spectrum of o-PP-5 exhibited other small signals, most likely derived from the other components present in the commercial MB.

### 2.4. Rheological Properties

Dynamic rheological experiments were performed to determine the processing behavior of the PP blends containing the two types of green colorants. Figure 5a,b shows the complex viscosity, *η**, measured at 230 °C, as a function of frequency for the PP loaded with transparent and opaque pigments, respectively. As seen, all samples exhibited a Newtonian plateau at low frequencies, followed by a strong shear thinning, i.e., a decrease in viscosity with frequency. Referring to the previous research, the inclusions of high-surface-area nano-additives, such as CNTs, even at low loadings, significantly increased the apparent viscosity of the PP polymer melts [29]. A certain increase in viscosity was also found in the iPP loaded with blue, white, and orange pigments [30]. In our case, the PP containing the opaque pigments exhibited an increasing *η** with their concentration when compared to the neat PP, while for their transparent counterparts, the effect on the *η** was rather marginal. To better interpret the differences and quantify the implications of green MBs, the viscosity data were plotted in the complex plane and fitted with the Cole–Cole equation, which is sensitive to structural changes in the polymeric materials and their (nano)composites [31]. The Cole–Cole expression for the complex viscosity, *η**, has the following form of Equation (2):(2)$$\eta^{*}\left(\omega \right)=\frac{\eta_{0}}{1+{\left(i\omega \lambda_{0}\right)}^{1-h}}$$
where *η*_0_ is the zero-shear viscosity, *ω* is the angular frequency, *λ*_0_ is the characteristic relaxation time, *h* is the dispersion parameter, and *i* is the imaginary unit (*i*^2^ = −1). The parameter *η*_0_ can be extracted through the extrapolation of the arc of the semi-circle on the real axis [32]. As seen in Figure 5c,d, the Cole–Cole model correlated well with the data that formed a semicircular shape, suggesting the absence of pigment/PP covalent bonding. Analysis of the parameters showed that the PP loaded with the transparent colorant exhibited *η*_0_, in the range of 2510–2677 Pa·s, which is comparable to the neat PP (2651 Pa·s). Interestingly, the presence of our transparent colorants, in some cases, slightly decreased the *η*_0_ value, which could be attributed to the excluded free volume in the vicinity of the pigment nanoparticles (NPs). A similar observation was reported by Jain et al. [33], who studied PP viscosity melts loaded with in situ prepared silica NPs. On the contrary, by adding the opaque pigments (o-PP-1, o-PP-3, and o-PP-5), the *η*_0_ parameter progressively increased, up to 2669, 2793, and 3034 Pa·s, respectively. The different manifestations were also observed for the relaxation time, *λ*_0_. Compared to the reference PP (*λ*_0_ of 0.229 s), the transparent pigments showed decreased *λ*_0_ values (0.184–0.191 s), which can be attributed to the intensified mobility of the PP chains [31], while the opaque pigments exhibited the opposite trend (0.232–0.252 s). Finally, the *h* parameter (referring to the distribution of molecular weights) ranged between 0.435 and 0.458 for all the samples, regardless of the type of pigments. Such *h* values are typical for polymers with a broad molecular weight distribution, such as PP or HDPE [34]. The molten-state rheological experiments suggested a different type of pigment/matrix interaction in the studied materials, which was further correlated with the solid-state mechanical behavior.

### 2.5. Mechanical Properties

The problem of low stiffness and strength values in polymer materials and their improvement by adding various inorganic particles to the base polymer material has been the subject of numerous studies [35,36,37,38,39]. Significant efforts have been made with (exfoliated) clays [40,41,42], carbon nanotubes [43,44], nanofibers with different aspect ratios [45], or ceramic particles [46], having at least one nanoscopic dimension [47]. In such reinforced nanocomposites, the mechanical properties are generally improved because the inorganic fillers possess much higher stiffness than the polymer matrix. However, very interesting effects occur in the case of mechanical strength, which is strongly dependent on the stress transfer between the particles and the matrix. For the particles with a good interfacial interaction, the stress can be effectively transferred within the nanocomposite, resulting in an increase in strength, while the opposite trend is usually observed for the systems with poor interfacial interactions [48].

The primary objective of this study was not to modify the mechanical properties of PP by adding inorganic fillers but to use the pigmented/dyed MBs to color the base material to achieve the desired shade and to identify the accompanying secondary effects occurring due to the presence of such additives. As mentioned above, in polymers, including semi-crystalline PP, the effect of inorganic particles, such as pigments. on the mechanical properties of the composite is significant because the presence of particles affects not only the composition of the composite, but also induces changes in the crystallinity [2,13]. As shown recently [49], the particle size also has a major influence on the mechanical strength of the nanocomposite. It was reported that the strength increases with decreasing particle size because the smaller nucleation nuclei produce a more heterogeneous structure of the crystals in the product. As seen in Figure 6a,b, the presence of green opaque colorant increased the UTS and modulus to a greater extent when compared to its transparent analog, which goes against the expected trends. The possible reason for such behavior is likely the enhanced interfacial adhesion in the PP loaded with the opaque colorant particles, positively affecting the UTS and the Young’s modulus [48]. This hypothesis is convincingly supported by the ductility data, displayed in Figure 6c. The elongation at break of the transparent products decreased by up to 16% when compared to the neat PP, which indicated deteriorated interfacial adhesion; this correlates well with the molten-state rheological observations (Figure 5). On the contrary, the PP loaded with the opaque colorant exhibited a significant increase in the elongation at break, by up to 266%, which clearly evidenced the superior interfacial adhesion. The results showed that even small amounts of the inorganic colorant additives, as low as 1 wt%, have a significant effect on the mechanical properties of the PP. The presence of the applied pigments, especially the opaque variant, has a positive effect on the mechanical properties of the resultant product.

When concerned with impact toughness, two theories can be distinguished. The first approach works with the evaluation of the notch toughness, *A*_k_, which is related to the crack size and fracture toughness. The second energy approach takes into consideration the critical energy, which is required to expand the crack by a unit area [50]. The process is also affected by interfacial adhesion, particle loading, and external factors, such as velocity of impact, temperature, etc. Figure 6d shows that the presence of the applied colorants had only a minimal effect on the *A*_k_ coefficient, regardless of the type of colorant and its concentration (within the investigated range). A similar trend was observed for the maximum force required to dynamically break the test specimen (Figure 6e). Thus, it can be concluded that the presence of the colored MB does not deteriorate the impact properties of the resultant PP products. The herein prepared colored PP formulations can be readily used in practical applications, as they have comparable or even slightly better mechanical properties than the widely used commercial PP.

### 2.6. Color Characteristics

The CIE color space coordinates *L**, *a**, and *b** were used to assess the color values, where *L** is a specific lightness parameter, parameter *a** defines colors on its axis from negative green to positive red, and parameter *b** defines colors on its axis from negative blue to positive yellow. The reflectance spectra and the optical parameters are shown in Figure 7 and Table 1, respectively. For transparent green pigment, there was a significant color change with the increasing concentration of the MB; parameter *a** increased in the negative direction from −3.37 to −15.99, and parameter *b** increased in the positive direction from 5.59 to 14.40. At the same time, parameter *L** decreased from 84.52 to 80.20. A higher concentration of this MB resulted in a richer and darker shade of green, closely reaching the RAL 6019 (pastel green) standard for t-PP-5. The evolution of optical properties was accompanied by changes in the reflectance spectrum (Figure 7a). For this reason, the dosing of the transparent MB is very important to achieve the desired shade. In contrast, for the opaque green MB, we observed very little color change when increasing its concentration in the PP matrix. The *L** and *b** parameters were almost identical, but the *a** parameter increased slightly from values of −47.58 to −42.23, indicating that there was a stabilization in the green hue direction with increasing concentration. The achieved color standard showed the most similarity with RAL 6037 and RAL 6038 (pure green and luminous green). The reflectance spectra were almost identical (Figure 7b), suggesting that lower concentrations of this MB are sufficient to achieve a stable green cover shade.

## 3. Materials and Methods

### 3.1. Materials

The commercial semi-crystalline PP (*α* isotactic, iPP, Mosten TB 003 Natural; MFI = 3.2 g at 230 °C/2.16 kg, M_W_ of 420,000 g/mol) was supplied from Unipetrol (Litvinov, Czech Republic). Two green colorants made by Gabriel-Chemie (Gumpoldskirchen, Austria) commercially branded as Maxithen were selected as the coloring substances. The first MB was PP616297 (containing transparent colorant concentrate), while the second one was PP620577/12 (containing opaque colorant concentrate). According to the literature, the transparent green is achieved by copper phthalocyanines, such as CI Pigment Green 7 [51], which can be referred to as an organic dye [4]. On the contrary, the opaque green color is achieved by the presence of inorganic pigments, such as cobalt oxides and chromium oxides, which are often combined with titanium oxide [27]. In both cases, the pigments are embedded in the same PP carrier (PPHD) at a typical concentration ranging from 5 to 50% [5]. Since both MBs are commercial products, the presence of the other compounds cannot be ruled out.

### 3.2. Mold Design

The injection mold was conceptualized following the EN ISO 294-4 standard [52], which describes the basic geometrical parameters and criteria, such as the length of the distribution system and the position and layout of the ejection system. A strong emphasis is paid to the location of the ejection system. However, some parameters are loose in this respect, including the arrangement of the product in the mold cavity and the design of the surrounding parts. For these reasons, the quality of the mold was inspected, and the injection molding process was simulated before employing our prototype mold (Figure 8) for the production of PP specimens.

### 3.3. Inspection of Cavity’s Dimensions

The main requirement posed to the injection mold is the dimension precision. The nominal value of test bodies specified in the aforementioned standard is 60 ± 0.1 mm in length and width and 2 ± 0.1 mm in thickness. The mold tolerances were ±0.05 mm for length dimensions and ±0.1 mm for thickness. An optical scanner, ATOS Triple Scan II (GOM, Mils, Austria), was calibrated and used to verify the dimensional accuracy of the mold. The MB170 scan volume was selected for the scanning procedure. The object was covered with the reference points used for the accumulation of the individual scans. No post-processing of the scan data was applied to evaluate the dimensions.

### 3.4. Formation of Basic Mixtures

The gravimetric technique was used to determine the exact amounts of the neat PP granules (as the base material) and each specific colorant (PP616297 and PP620577/12) to prepare the polymer blends. The fabricated formulations consisted of MB concentrations of 1, 3, and 5 wt%. These compositions were selected since this concentration range is widely used in practical applications from a technological and economic point of view.

### 3.5. Fabrication of the Samples

The specimens were injection-molded using the Allrounder 470 H device (Arburg, Lossburg, Germany). The injection mold was selected according to the EN ISO 294-4 standard, which is suitable for the determination of the shrinkage of thermoplastic materials. The processing parameters were selected based on the material data sheet: the temperature of the injection mold was set to 40 °C, with a cooling time of 30 s, a dose length of 24 mm, a holding pressure switchover of 8 mm of the dose length, an injection pressure of 80 MPa, an injection speed of 40 mm/s, a holding pressure of 70 MPa, a holding pressure duration of 10 s, a melt temperature range from 220 to 240 °C, and a cycle duration of 50 s. These conditions were kept constant for both coloring MBs, regardless of their concentration. In addition to the reference (neat PP), the PP blends containing 1, 3, and 5 wt% of PP616297 (transparent; t) and PP620577/12 (opaque; o) colorants were investigated. The sample IDs were as follows: neat PP, t-PP-X, and o-PP-X, where X denotes the concentration of each colorant in wt%.

### 3.6. Shrinkage Determination

The shrinkage measurement was performed on a Contura Zeiss G2 (Zeiss, Oberkochen, Germany) equipped with a touch probe of 1 mm in diameter, specifically for its high precision, high reproducibility, and partial automation of measurement. The accuracy of the instrument (as given by the manufacturer) is 1.8 µm on the nominal length of 300 mm. The measurements were taken at points only, in accordance with the standard (EN ISO 294-4), and the results were evaluated by Calypso software (version 6.2).

The specimens were separated from the injection system immediately after the injection in order to avoid any distortions in shrinkage measurements. The results were gained after 7 days, during which time the samples were kept under laboratory conditions (temperature of 21 ± 2 °C, humidity of 55–60%). Each dimension of each sample (neat PP, t-PP-1–5, and o-PP-1–5) was measured 15 times. The overall shrinkage was recorded in both, the flow direction, *S*_TF_, and the transverse direction, *S*_TT_, and was calculated using the following Equations (3) and (4) given by the EN ISO 294-4 standard:(3)$$S_{\mathrm{TF}}\left[\%\right]=\frac{L_{M}-L_{S}}{L_{M}}\times 100\;\left[\%\right]$$
(4)$$S_{\mathrm{TT}}\left[\%\right]=\frac{b_{M}-b_{S}}{b_{M}}\times 100\;\left[\%\right]$$
where *L*_M_ and *L*_S_ denote the main dimension (length) of the mold and the sample, respectively, while *b*_M_ and *b*_S_ denote the secondary dimension (width) of the mold and the sample, respectively. Figure 9 shows the testing assembly for measuring the dimensional stability of the specimens.

### 3.7. X-ray Diffraction Analysis

The changes in the crystalline structure were examined by wide-angle X-ray diffraction (XRD) analysis. The diffraction patterns were obtained using an X-pert Pro XRD system (PA Nalytical, Almelo, The Netherlands) using the Ni-filtered Cu K*α* radiation source (λ = 1.5406 Å) in the transmission mode. The XRD device was operating within a 2θ range of 5–30° at a scan speed of 4.5°/min. The investigation was carried out at laboratory temperature. The XRD data were collected in the middle of each specimen from an area sized 1.5 cm × 0.5 cm.

### 3.8. Scanning Electron Microscopy and Energy-Dispersive X-ray Spectrometry

Prior to the analysis, the samples were sputtered with a thin layer of gold using the SC760 coating device (Quorum Technologies, Lewes, UK) to avoid the charging effects. The dimensions of pigments and their distribution inside the PP matrix were studied on the freeze-fractured (liquid N_2_) samples using scanning electron microscopy (SEM). The observations were performed on the field-emission SEM device (Nova NanoSEM 450, FEI, Hillsboro, USA). Upon insertion of the Octane Plus detector, the SEM device was also used for energy-dispersive X-ray spectroscopy (EDX) to analyze the elemental compositions. Depending on the detector, the device was operated at an accelerating voltage of 5–18 kV with a spot size of 3.0–4.5.

### 3.9. Melt Rheology

The rheological behavior of the molten state samples was investigated on a modular rheometer, the Physica MCR502 (Anton Paar, Graz, Austria), equipped with a TC 30 temperature controller and a CTD600 heating chamber. The parallel-plate configuration (PP25) with a diameter of 25 mm was used, while the gap was set to 1.8–2.0 mm. The dynamic frequency sweeps were performed at 230 °C to simulate the processing conditions (Section 3.5), within a frequency range from 0.01 to 50 Hz, at a strain amplitude of 0.5%, lying within the linear viscoelastic (LVE) region. To avoid thermal oxidation, the chamber was purged with nitrogen.

### 3.10. Tensile Properties

The determination of the tensile properties was carried out according to the technical standard EN ISO 527-1. The samples were fabricated using an Allrounder 470 H (Arburg, Lossburg, Germany) with a crossbar speed rate of 50 mm/min. The tensile data were used to determine the following properties: Young’s modulus, *E*; ultimate tensile strength, UTS; relative elongation, *ε*. Herein, the UTS parameter was considered the most significant due to its greatest importance from a practical point of view.

### 3.11. Impact Toughness

The determination of impact toughness was performed by the Charpy method in accordance with EN ISO 179-1. The characterization was carried out on a pendulum impact tester, the Zwick HIT50P (Zwick Roell, Ulm, Germany), using specimens with a notch depth of 2 mm. Two parameters were evaluated, namely the notch toughness factor, *A*_k_, and the force required to break the sample, *F*.

### 3.12. Colorimetric Measurements

The colorimetric properties were analyzed on an UltraScan PRO spectrophotometer (HunterLab, Reston, VA, USA). Illumination was provided by a pulsed xenon lamp using the CIE standard illuminant D65 white lighting. The colorimetry detector consisted of three sensors that corresponded with the cones of the human eye. The light reflected from the sample was measured with a diffuse illumination geometry and a 10° viewing angle. The reflectance spectra of the samples were measured in a wavelength region from 350 to 750 nm with a spectral resolution of 5 nm. To ensure accuracy, the measurement was conducted 3 times on 10 individual samples for each system. Because of minimal variations, the resulting optical values were averaged for a clearer interpretation. The white calibration plate using a D65 illuminant was performed prior to each sample group analysis. The analyzed area was 10 mm^2^. The results were inverted using the CIE lab parameter.

## 4. Conclusions

Our study reveals key findings related to the influence of green transparent and opaque commercial MBs on the behavior of isotactic PP. The dyed/pigmented samples were injection-molded using a custom-built mold constructed according to ISO standards. The shrinkage of PP samples increased in line with MB concentrations but was more pronounced when using opaque MB. It was found that opaque MB induced a higher crystallinity (by 10.6%) and provided superior mechanical properties, including a 38% increase in Young’s modulus and a 16% increase in UTS, when compared to neat PP. Transparent MB, on the other hand, did not show such a significant increase, suggesting a lower nucleating capability of dyes. SEM/EDX analysis confirmed a good dispersion of pigments and revealed the presence of the elements Cr, Cu, and Ti, commonly used in green coloring substances. We also observed an increase in the *η*_0_ values of molten-state PP when using opaque MB, indicating hindered mobility of polymer chains in the presence of the pigment (nano)particles. It was found that the addition of MB caused dramatic changes in the CIE color space, which had desired implications for the visual characteristics of the material. The optical properties progressively evolved with the loading of transparent MB, closely reaching the RAL 6019 standard. On the contrary, the color was saturated for the opaque MB while resembling RAL 6037 and RAL 6038 standards; thus, dosing this MB beyond 1 wt% did not yield a significant difference. Our study showed that the tested green PP formulations can be used in practical applications as they have comparable or even better mechanical properties than the basic PP, but in applications where dimensional stability is emphasized, the shrinkage phenomenon must be considered.

## Figures and Tables

**Figure 1 ijms-24-09924-f001:**
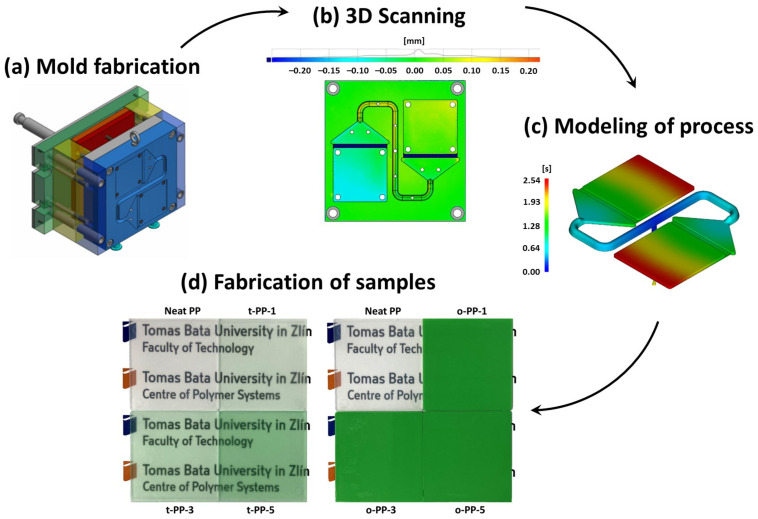
A brief overview of the research protocol. The design (**a**) and 3D scanning (**b**) of the mold, with the bar map showing the deviations from the nominal values. The simulation of the mold filling process (**c**) with the bar map showing the filling time. The digital photographs of the produced samples (**d**); when placed on a flat surface, the samples showed remarkable differences in transparency.

**Figure 2 ijms-24-09924-f002:**
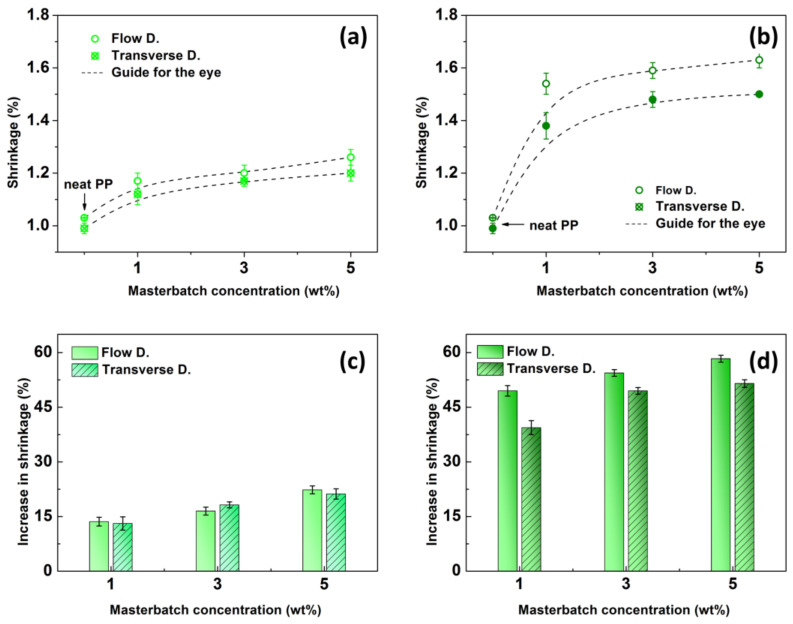
Dependence of the shrinkage on the MB concentration for the transparent PP (**a**) and its opaque analog (**b**), and the relative increase in shrinkage for the transparent PP (**c**) and its opaque analog (**d**).

**Figure 3 ijms-24-09924-f003:**
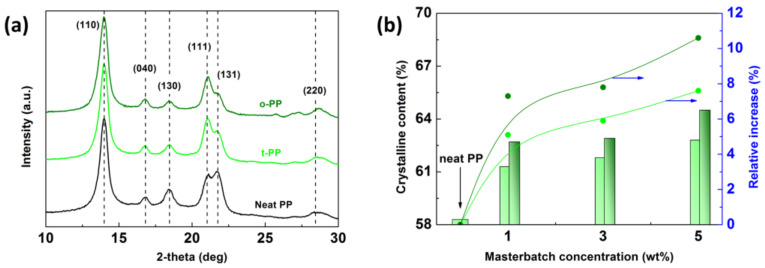
XRD patterns for the neat PP and its analogs containing transparent and opaque MB (5 wt%) (**a**), and the dependence of the crystalline phase content and its relative increase in relation to the transparent/opaque (light/dark green) MB concentration (**b**). Blue arrows in section (**b**) relate to the right-hand axis.

**Figure 4 ijms-24-09924-f004:**
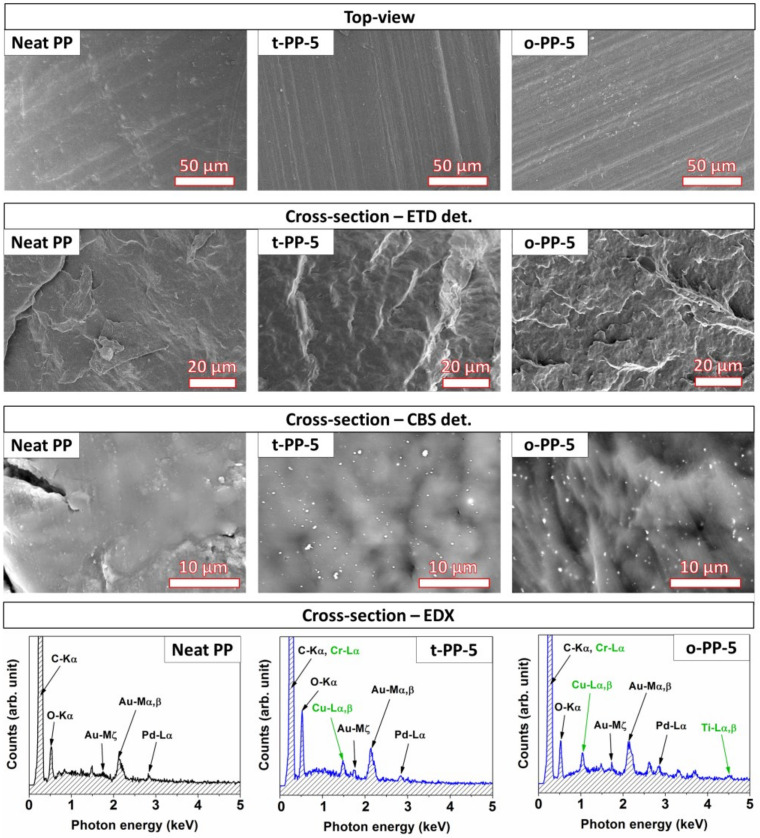
SEM micrographs of the surfaces, cross-sections, and magnified views of regions of the neat PP, transparent PP, and opaque PP; the corresponding EDX spectra (taken from the areas of 50 µm × 50 µm).

**Figure 5 ijms-24-09924-f005:**
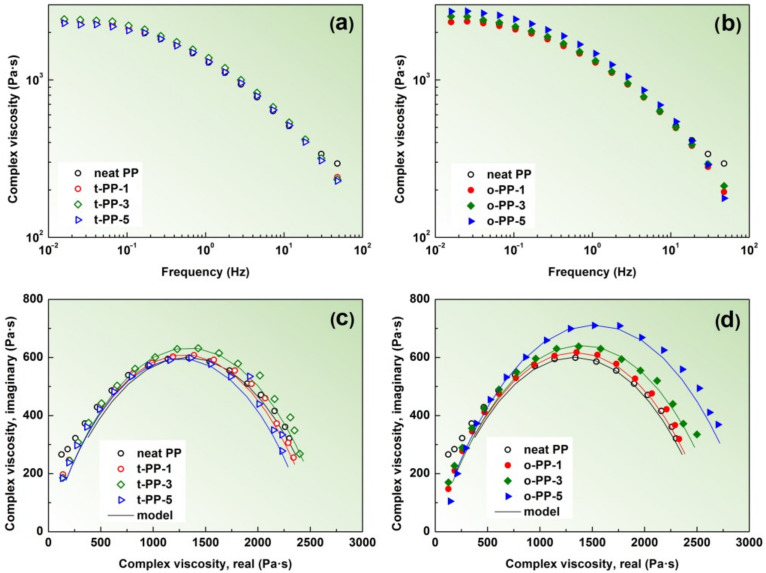
The rheological properties of the molten-state samples. The complex viscosity as a function of frequency for transparent PP (**a**) and opaque PP (**b**); the corresponding Cole–Cole plots (**c**,**d**).

**Figure 6 ijms-24-09924-f006:**
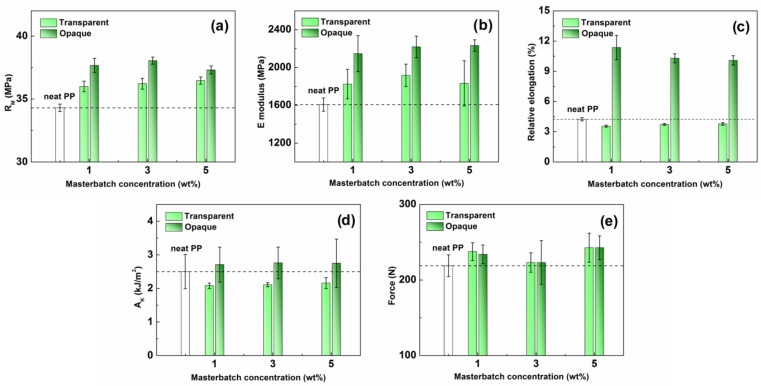
Dependence of the UTS (**a**), Young’s modulus (**b**), relative elongation (**c**), notch toughness factor (**d**), and maximum impact force (**e**) on the MB concentration for the transparent PP and its opaque analog. The dashed lines represent the values of the neat PP.

**Figure 7 ijms-24-09924-f007:**
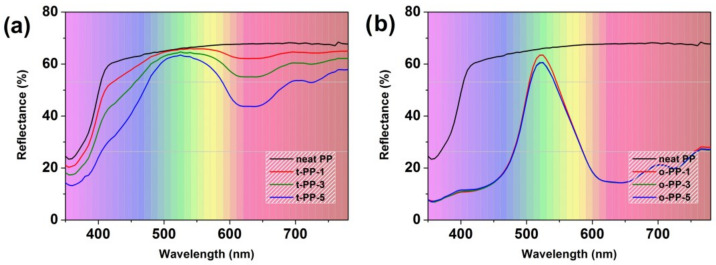
Reflectance spectra of samples colored with transparent green MB (PP616297) (**a**) and opaque green MB (PP620577/12) (**b**).

**Figure 8 ijms-24-09924-f008:**
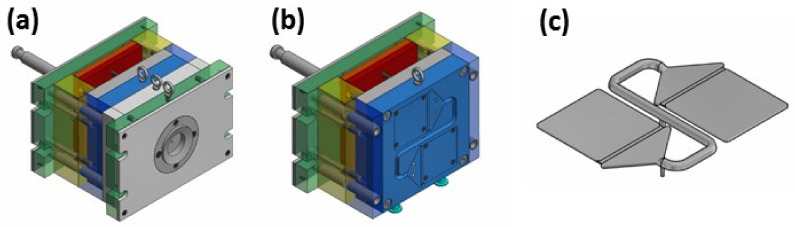
The concept of the injection mold for the production of test bodies. The overall injection mold assembly (**a**), the mold cavity including the delivery system (**b**), and a model of the final product (**c**).

**Figure 9 ijms-24-09924-f009:**
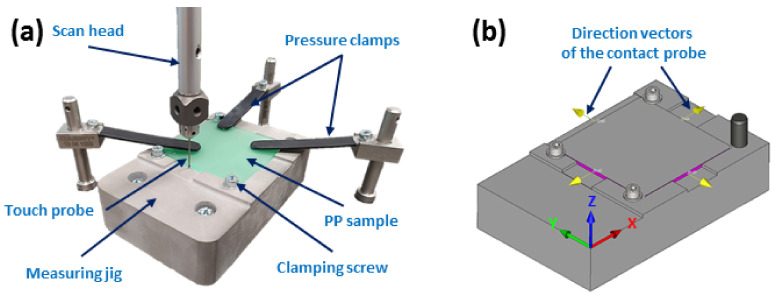
The assembly for the measurement of the dimensions for the specimens. The actual measurement waveform (**a**); the directional displacement vectors of the touch probe in the automatic mode (**b**).

**Table 1 ijms-24-09924-t001:** Quantifying the sample in the CIE lab color space and by hexadecimal coding.

Sample ID	D65/10°			Color (Hex)
	*L**	*a**	*b**	
Neat PP	85.20	−0.20	3.41	#F5F4F3
t-PP-1	84.52	−3.37	5.59	#E7EAE3
t-PP-3	82.87	−8.46	9.24	#E1E9DE
t-PP-5	80.20	−15.99	14.40	#D8E7D5
o-PP-1	68.91	−47.58	33.92	#68C87E
o-PP-3	68.40	−46.58	33.40	#60C375
o-PP-5	68.35	−42.23	33.26	#5DBA67

## Data Availability

The data will be available from the corresponding author following reasonable request.

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
