# Peer review of "Injection-Molded Isotactic Polypropylene Colored with Green Transparent and Opaque Pigments"

_ijms, 2023, doi:10.3390/ijms24129924_

Round 1
Reviewer 1 Report
This paper is reporting the nucleation effect and subsequent mechanical performance of polypropylene containing green pigments. I am afraid it is missing some key aspects and the impact would be low both from academic and applications point of view, thus I am afraid I cannot recommend publication.
Starting from the title, it is not reflecting the content and purpose very accurately in my opinion, as also the authors in lines 397-400 state: "The primary objective of this study was not to modify the mechanical properties of PP by adding the inorganic fillers, but to use the inorganic pigments to color the base material to achieve the desired shade, and to identify the accompanying secondary effects occurring due to the presence of such additives". Given the vast literaure on green pigments and other nucleating agents and nanocomposites, it is hard to see why this work should stand out or even how it compares to the present state of the art.
Nevertheless, in lines 19, 417 and 480, a comparison to neat PP is made to conclude an improvement of 266% in elongation at break. This is like comparing apples with pears, as obviously nucleated PP is much different in that perspective. Of higher usefulness would be to compare with other reference green pigments, which is missing in this work. Such huge % differences are usually seen very sceptically! At the same time, in the abstract, the description of results is made with vague expressions such as "slightly increased", "minimally affected", "the effect [...] was significant". Exact numerical data would help the reader more.
The term "nanocomposites" is used to describe the compounds. No size information on the pigment fillers is given in section 2.1 and the only results from which their size can be estimated are the SEM images of Figure 6. The smallest of the white spots (pigment agglomerates) would be about 200 nm, which is larger than the conventional definition of nano (<100 nm in at least one dimension). How would it be demonstrated that the distribution is in a sub 100 nm level?
Also on the raw materials of Section 2.1:
What are some basic molecular properties of the polypropylene, such as Mw, MWD, MFR?
What is approximately the percentage of the green pigments in the whole PP/pigment masterbatch? (important for evaluating the nucleating effects at the end of the day).
Figure 1: As the specimens were practically produced according to the standard EN ISO 294-4, the figure is not providing any additional valuable information.
Lines 235-348, Figure 3 and Graphical Abstract: Modelling of the system is mentioned and complicated images are shown, but the results are not discussed in appropriate depth. Seems rather not adding much to the total discussion; I would suggest removing.
The manuscript is about green pigments, but I feel that section 3.6 discussing the actual colour effectiveness is too short. Also according to the authors (lines 103-104) "The aim of this research is to complexly investigate the behavior of PP upon introducing two different color pigments in various concentrations.". For example, what was the intended green reference standard (RAL etc) that was aimed, and how close to that was the resulted shades?
Figure 4: Panels c and d are not conveying additional information compared to panels a and b, and can be omitted in my opinion.
Other points:
Graphical Abstract: The text ("Tomas Bata University...") is too small to read. Here not as important as in Figure 3, but still the reader would think that important information is there to read.
Equations: There is something with the numbering of the equations. Starting with the first two in pages 4 and 5, where they meant to be 1a and 1b? Then the equation in page 8 would be #3 and the one in page 11 #4?
Line 33: is the injection molding --> is injection molding
Lines 206, 344: 230°C --> 230 °C
Line 251: modelling --> modeling (for uniformity with US English like elsewhere in the manuscript)
Author Response
Dear Reviewer, our responses are enclosed in the attached file.

Reviewer 2 Report
The manuscript by Cvez and colleagues refers to some effects of pigments on the mechanical properties of PP. The results are interesting but must be presented and discussed taking into account the comments/suggestions reported below.
- The use of the term nanocomposite for the PP coloured with organic dyes as phtalocyanine green (called t-PP) is not appropriate whereas the o-PP prepared with an inorganic pigment may be referred as PP composite. As a consequence, the term cannot be used in the title and anywhere else along the manuscript, as well as avoiding direct references to nanocomposites in the manuscript.
- The pigments/dyes must be perfectly characterized before mentioning their effects on the PP, eventually even already in the materials and methods section.
- The first paragraph of the introduction “introduce” very well know information which may be found in encyclopedias of textbooks and only at the end center on the specific topic of the manuscript. The effect of pigments/dyes on the crystallization of polymers and specifically of PP is a very well know topic, both industrially and academically. As a fact is possible to find reference works much earlier than the few mentioned in the text, also citing, as an example, the effect of phtalocyanine, among the others on the mechanical properties of PP. My suggestion is to rewrite the introduction mentioning what is already well known in the sector (specifically including all the studies on the mechanical properties changes, also in the industrial practice, focusing on what this contribution will further clarify.
From section 3 on, I only suggest the following:
- Lines 317-328 the concept are obvious.
- 3.6 I do not know whether also this information is obvious to anyone measuring colors
Conclusions must point out what this manuscript revealed, which are the main differences between the results discussed in section 3 and what was already known. It cannot be a resume/repetition of what the reader just read.
It must be improved
Author Response

(The authors gave the same response as above.)

Round 2
Reviewer 1 Report
I would like to thank the authors for the thorough review of both the text and the figures of the manuscript. Substantial improvement with solid justification is achieved; I would certainly support publication.
Reviewer 2 Report
It may accepted as it is
some minor editing is still required